# Experimental Study on Vibration Characteristics of Unit-Plate Ballastless Track Systems Laid on Long-Span Bridges Using Full-Scale Test Rigs

**DOI:** 10.3390/s20061744

**Published:** 2020-03-20

**Authors:** Weiqi Zheng, Xingwang Sheng, Zhihui Zhu, Tianjing Luo, Zecheng Liu

**Affiliations:** 1School of Civil Engineering, Central South University, Changsha 410075, China; wqzheng@csu.edu.cn (W.Z.); zzhh0703@163.com (Z.Z.); xwsheng@csu.edu.cn (T.L.); zclcsu@163.com (Z.L.); 2National Engineering Laboratory for High Speed Railway Construction, Changsha 410075, China; 3China Railway Engineering Consulting Group Co., Ltd., Beijing 100020, China

**Keywords:** high-speed railway, unit-plate ballastless track, hammering test, full-scale test rig, vibration transmission, vibration characteristic, time domain, frequency domain

## Abstract

In this work, we present a series of hammering tests on full-scale unit-plate ballastless tracks used for long-span bridges. There is no denying that it is a new attempt to pave ballastless tracks on high-speed railway long-span bridges; the related issues deserve to be studied, and especially the vibration characteristics. Hence, the vibration characteristics and transmission rules of the ballastless track with geotextile or rubber isolation layers are explored, and the vibration reduction effect of the rubber isolation layer is analyzed. The main conclusions are as follows: the isolation layers change vibration modes and transmission characteristics of ballastless tracks; the introduction of the rubber isolation layer makes the excited vibration frequency range of the ballastless track concentrated; and the vibrations of the ballastless track with the rubber isolation layers are stable. Moreover, the rubber isolation layer has an obvious attenuation effect on vibration transmission in ballastless track structures. When the vibration is transmitted from the rail to the bridge deck, the vibration level differences of the ballastless track with rubber isolation layers are 20 dB larger than that of the ballastless track with the geotextile isolation layers. The vibration attenuation rate of the rubber isolation layer is about ten times larger than that of geotextile isolation layer.

## 1. Introduction

Nowadays, high-speed railways have developed rapidly around the world, especially in China [1]. High-speed railways have proven to be one of the most significant technological advances in recent decades. As one of the commonly used track types, the ballastless track has been widely employed in high-speed railways. However, it has also given rise to some problematic issues that must be addressed. One particular concern is related to the comprised noise and vibrations induced by the passage of high-speed trains [2], which has been a disturbance to the people who live or work near the railways. Therefore, increasing requirements of noise reduction and vibration attenuation for high-speed railways should be put forward to avoid environmental disturbance.

Based on this, many authors have carried out some experimental measurements and numerical simulations of train-induced vibrations in recent years [3,4,5,6,7]. Cox et al. studied the influence of fastener stiffness on dynamic properties of three different floating track slabs [8]. Zhao et al. investigated that damping pad can effectively reduce the vibration of bridge structure, but the vibration for track slab and rail will be enlarged [9,10]. Lee et al. established the structural model of a viaduct, and then its vibration responses and noise radiation characteristics were obtained [11]. Kraśkiewicz et al. studied the static and dynamic performance of the rubber isolation layer and put forward the relevant parameters for the effectiveness of track vibration reduction according to German standard [12]. Li et al. took the 32 m pre-stressed concrete simple box girder in high-speed railway as a research object, the vibration transmission characteristics of the railway concrete simple box girder were discussed based on a series of hammering tests [13].

However, systematic studies on the vibration characteristics of the unit-plate ballastless track systems are still quite limited among railway researchers. The types of unit-plate ballastless tracks are usually paved on high-speed railway long-span bridges, and some new components (such as new type isolation layer) are employed in ballastless tracks to improve their mechanical behaviors [14]. As is well known, the conventional unit-plate ballastless track consisted of rails, fastener systems, track slab, self-compacting concrete layer, isolation layer, and basement (Figure 1). Setting damping materials, including rail pads and elastic isolation layers, in these ballastless tracks is an effective way to mitigate vibration and noise at the source [15]. Besides this, the introduction of an elastic isolation layer can separate the self-compacting concrete and basement as well as effectively absorb the vibrations in comparison with the conventional ballastless track systems [16].

At present, model testing is always used to investigate the dynamic characteristics of railway tracks [17,18] and using a full-scale model can effectively reduce the uncertainties in experimental measurements [19]. In this work, a set of full-scale test rigs were constructed based on two types of ballastless tracks employed in China high-speed railways, which are characterized by their isolation layer types. These tests aim to reveal the vibration characteristics of the unit-plate ballastless tracks used for long-span bridges. Hereafter, the modal characteristics of the unit-plate ballastless tracks with different types of isolation layers are investigated, and then the dynamic characteristics and vibration transmission characteristics of these unit-plate ballastless tracks are also evaluated and discussed in detail.

## 2. Test Program

### 2.1. Description of the Test Rig

In China, the unit-plate ballastless track mainly includes CRTS I type double-block ballastless track, CRTS III type ballastless track, etc. [20,21,22]. This work took CRTS I type double-block ballastless track as an example to carry out experimental research. A series of full-scale CRTS I type double-block ballastless tracks were laid on the existing simply supported steel-concrete composite-girder in the laboratory. These ballastless tracks with different types of isolation layers were arranged into two lines on the composite-girder and the composite-girder had already been built up based on a long-span cable-stayed bridge. In this work, these unit-plate ballastless tracks were fabricated to pave on long-span bridges, and they were arranged with geotextile or rubber isolation layers for further research. The composite-girder with these full-scale unit-plate ballastless tracks can be seen in Figure 2.

The CRTS I type double-block ballastless track was first introduced and developed by Rheda (2000) and was re-innovated based on China’s actual situation. Additionally, it has already been used in Wu-Guang and He-Fu passenger dedicated lines in China. The length and width of this type of unit-plate ballastless tracks are 5920 and 2800 mm. The thickness of the concrete track slab and basement are 260 and 240 mm. Both the basement and track slab are made of C40 concrete, and CHN60 rail (similar to UIC60 rail) and WJ7-B constant resistance fasteners are employed in these ballastless track systems [23].

More important, the geotextile isolation layer has been widely used in the construction of CRTS I type double-block ballastless track. The thickness of the geotextile layer is always 4 mm, and it meets all the requirements in China standard [24]. The rubber isolation layer used in this work is a kind of the commonly used under-ballasted mat, which has been already introduced into high-speed railways. The thickness of this rubber isolation layer is 27 mm, which is composed of two different layers of rubber structures. Moreover, the compressive stiffness of this rubber isolation layer is 0.025 N/mm^3^. More detailed information about the isolation layers can be found in Ref [25]. The unit-plate ballastless tracks with different types of isolation layers can be seen in Figure 3.

### 2.2. Sensors and Test Procedure

In this work, the modal characteristics, dynamic behaviors, and vibration transmission of the unit-plate ballastless tracks were investigated by conducting a series of hammering tests. The hammering test, which is to apply a pulse excitation to the structure, is a test on the structure in the frequency range to be studied [26]. More detailed information about this method can be found in Reference [27]. This test entails producing a transient excitation on the rail, and recording vibration values of ballastless track systems response to such a load. Therefore, the vibration characteristics of the rails and ballastless tracks can be obtained, the vibration transmission of the components in the ballastless track systems and the vibration reduction of the rubber isolation layers can be studied. Here, the excitation hammer equipment adopted the DFC-1 hammer device. The test equipment included the DASP system, INV3018CT acquisition instrument, and INV9822/9828 acceleration sensor. The sensors and hammering test can be seen in Figure 4.

In the modal test, four positions on the concrete track slab were randomly selected for excitation. Additionally, in the vibration transmission characteristic test, the excitation points were on the top surface of rails, and five repeated hammering tests were carried out in each working condition, labeled as Test #1 to Test #5.

### 2.3. Modal Measurement

In modal measurement procedure, a series of measuring points were evenly arranged on the top surface of concrete track slab, and an acceleration sensor was installed on each measuring point to obtain vertical acceleration. The layout of these measuring points is shown in Figure 5.

### 2.4. Vibration Characteristics Measurement

In this work, the vertical responses of the unit-plate ballastless tracks with different isolation layers were measured to research the vibration characteristics. Two test conditions were organized and carried out, labeled as Condition #1 and Condition #2, respectively. Condition #1: impacting rail at central position of the ballastless track with geotextile isolation layers; Condition #2: impacting rail at central position of the ballastless track with rubber isolation layers. The vibration characteristics of different ballastless tracks were studied by symmetrical arrangement of measuring sensors in these two working conditions. The arrangements of measuring sensors are shown in Figure 6, and each measuring sensor has its own unique label.

## 3. Modal Characteristics

The modal characteristics of the track slab in these two types of ballastless tracks were obtained according to the test data of hammering tests. The first three orders of modal modes of the track slab are shown in Table 1.

It can be seen from Table 1 that for the ballastless tracks with different types of isolation layers, the modal modes and vibration characteristics of the track slabs are different. As the rubber isolation layer is much more elastic than the geotextile isolation layer, the track slab on the rubber isolation layer is similar to that laid on elastic foundation. The decrease of foundation stiffness leads to a decrease in vibration frequency of the track slab, which also has a certain impact on its vibration mode characteristics.

## 4. Vibration Characteristics

### 4.1. Vibration Characteristics of Rails

The time-history vibration curves of the rails can be obtained by the acceleration sensors installed in this work, and then the frequency-history vibration curves can be obtained by the Fast Fourier Transform technique. Based on this, the time-domain and frequency-domain results of some measuring sensors on rails in these two conditions are shown in Figure 7.

It can be seen from Figure 7 that the vibration peak values of sensors R1F1 and R1J1 are reached rapidly, and the vibration amplitude basically disappears within 0.03 s. In the time domain, the rail vibration shows obvious multi-peak characteristics in Condition #1. Moreover, the vibration peak value of the rail decreased stably after maintaining a certain time in Condition #2. In the frequency domain, the rail is obviously excited at most of frequency ranges in Condition #1, of which the frequency range of 2000 to 3000 Hz is the largest. The excitation frequency range of the rail is relatively concentrated in Condition #2, of which the frequency about 2500 Hz is the largest. Due to the different types of isolation layers underneath the track slabs, there are obvious differences in time-domain and frequency-domain vibration characteristics of the rails. Laying rubber isolation layer underneath the track slab makes the vibration frequency of the rail centralized, and the vibration attenuation becomes stable.

### 4.2. Vibration Characteristics of Track Slabs

The time-domain and frequency-domain results of some measuring sensors on concrete track slabs in these two conditions are shown in Figure 8.

As shown in Figure 8, in the time domain, the vibration attenuation of the concrete track slab is similar to that of the rail. Moreover, the vibration of the concrete track slab presents steady attenuation characteristics. In the frequency domain, the vibration laws of the ballastless track in these two working conditions are different. The excitation frequency range of sensor S1J1 in Condition #2 is more concentrated, and mainly concentrated at about 2000 Hz, while the excitation frequency range of sensor S1F1 in Condition #1 has a wider range, and there is little difference in the frequency range of 1000 to 2000 Hz.

## 5. Vibration Transmission Characteristics

### 5.1. Evaluation Method of Vibration Transmission

Based on the vibration characteristics of the ballastless tracks, the vibration transmission laws of the track slab in these ballastless tracks are studied. In this work, the virtual vibration values of each measuring sensor on ballastless track structures are compared and analyzed to study the vibration transmission and reduction characteristics. The virtual vibration value $\;a_{rms}\;$ can be expressed by the following Formula (1) [28]:
(1)$$a_{rms}=\sqrt{\frac{1}{∆t}\overset{t_{2}}{\underset{t_{1}}{\int }}{\left[a\left(t\right)\right]}^{2}dt}$$

In Formula (1): a(t) is the acceleration function with time t as the variable; ∆t is the total time length; $t_{1\;}$ is the start time of integration; $t_{2}\;$ is the end time of integration, with taking the vibration acceleration falling to 1% of its peak value as the endpoint.

Since the vibration attenuation is not apparent in the longitudinal and transverse direction of the ballastless track, the attenuation rate is expressed in the form of percentage; while in the vertical direction of the ballastless track, the acceleration of each layer in the ballastless track has a large attenuation, so it is more reasonable to describe the vertical attenuation rate by using the vibration acceleration level differences.

At present, the vibration acceleration level is indicated by the following Formula (2) [28]:


(2)$$V_{AL}=20\log\left(\frac{a}{a_{0}}\right)\left(dB\right)$$


In Formula (2), $a_{0}$ is the reference acceleration. Generally, it is taken as $\;10^{-6}\;m/s^{2}$.

Then the acceleration level difference between A and B can be expressed as the following Formula (3):
(3)$$V_{AL}=V_{AL}\left(B\right)-V_{AL}\left(A\right)=20\log\left(\frac{a\left(B\right)}{a\left(A\right)}\right)\left(dB\right)$$

Therefore, the attenuation rates and the level differences of the vibrations between different measuring points are taken as the evaluation indexes to investigate the vibration characteristics and transmission law of the two types of ballastless tracks in the longitudinal, transverse and vertical directions in these two conditions. Additionally, the vibration acceleration index of other measuring points is compared with that of the base point, which is next to the exciting location.

### 5.2. Vibration Transmission Characteristics of Rails

The acceleration values of the measuring sensors on rails in Condition #1 and Condition #2 are compared and analyzed. Based on the virtual values of sensors R1F1 and R1J1 in these two conditions, the attenuation rates of other measuring points relative to the test results of sensors R1F1 and R1J1 are calculated, respectively. The longitudinal vibration transfer characteristics of rails are shown in Figure 9.

As shown in Figure 9, in Condition #1 the longitudinal attenuations of rail vibration virtual values are characterized by fluctuation. The attenuation amplitudes of sensors R1F2 and R1F4 are relatively large, while that of sensor R1F3 is relatively small. In Condition #2, the rail vibration virtual values basically decreased with the distance to the rail excitation point increasing, and there is no obvious fluctuation change founded. It can be considered that different types of isolation layers underneath the track slabs have a certain impact on the rail vibrations. For the ballastless track with geotextile isolation layer, the rail vibration is restrained at fasteners. However, for the ballastless track with rubber isolation layer, the restraint effect of fasteners on rail vibration is not apparent.

The test data of these sensors on rails in the two conditions are selected to study the longitudinal transmission rules of rail vibration in frequency domain, as shown in Figure 10.

Figure 10 shows that regardless of the types of isolation layers, there are no obvious differences in rail vibration in the frequency range of 1000 to 2000 Hz in the process of rail vibration transferring from sensor R1F1 (or R1J1) to sensor R1F5 (or R1J5). However, in the frequency range of 3000 to 5000 Hz, the vibration level differences changed significantly, and the vibration loss is the largest.

For the ballastless tracks with different types of isolation layers in this work, the variations of acceleration level differences of the rail in frequency range of 0 to 1000 Hz are similar. Due to the existence of the rubber isolation layer, the acceleration level difference in Condition #2 is about 20 dB larger than that in Condition #1, which means that rubber isolation layer has a better damping effect. Moreover, there are significant differences in the acceleration level differences between the two types of ballastless tracks in frequency range of 2000 to 3000 Hz, which shows that this frequency range is sensitive to the vibration reduction and isolation effect of the rubber isolation layer.

### 5.3. Vibration Transmission Characteristics of Track Slabs

The test data of the sensors on the track slabs in Condition #1 and Condition #2 are selected for further research. Based on the virtual values of sensors S1F1 and S1J1 in the two conditions, the attenuation rates of other measuring points are calculated. The vibration transfer rules of the track slab are shown in Figure 11.

It can be seen from Figure 11 that in Condition #1, the virtual value of vibration acceleration decreases by 30% in the process of vibration transmission to track slab. Meanwhile, the virtual value of vibration decreases by 60% in Condition #2. It shows that the rubber isolation layer has an obvious attenuation effect on vibration transmission of track slabs.

Moreover, the test data of the sensors on track slab in Condition #1 and Condition #2 are selected, and then the vibration level differences of other measuring sensors are calculated based on the vibration acceleration level of sensors S1F1 and S1J1, so as to study the vibration transmission laws of the concrete track slabs, as shown in Figure 12.

As can be seen from Figure 12, for Condition #1 and Condition #2, the variation of vibration level differences in frequency ranges of 0 to 1500 Hz and 3500 to 5000 Hz are similar, and the variations of vibration level differences in frequency range of 1500 to 3500 Hz are obvious. The level difference in frequency range of 1500 to 3500 Hz in Condition #2 is about 15 dB larger than that in Condition #1. It can be seen that the vibration of track slab in the frequency range of 1500 to 3500 Hz is obviously attenuated due to the existence of the rubber isolation layers.

### 5.4. Vibration Isolation Performances

The vibration data of different structural layers in these two ballastless tracks in Condition #1 and Condition #2 are selected to study the vertical vibration transmission law. Based on the virtual values of vibration acceleration of sensors R1F1 and R1J1, the vibration level differences of other sensors are calculated, and then the vibration reduction and isolation effect of different types of isolation layers in ballastless tracks are obtained, as shown in Figure 13.

It can be seen from Figure 13 that the vibration transmitted from the rail to the bridge structure shows a general attenuation trend. The vibration levels of the concrete bridge deck and the steel box floor are mainly equivalent to the environmental vibration level. In Condition #1, the virtual value of the vibration attenuates about 60 dB from rail to bridge deck. However, in Condition #2, the virtual value of the vibration attenuates about 80 dB from rail to bridge deck. Compared with the geotextile isolation layer, the rubber isolation layer has an obvious damping effect, and the damping effect of the rubber isolation layer is about ten times that of the geotextile isolation layer.

Furthermore, the vibration level differences of sensors S1F1 and S1J1 on track slabs are calculated based on the vibration level of sensors R1F1 and R1J1, and the vibration level differences of sensor Q1 on bridge deck are calculated based on the vibration levels of sensors S1F1 and S1J1, so as to study the vertical vibration transmission laws in ballastless tracks with geotextile isolation layers or rubber isolation layers in frequency domain, as shown in Figure 14.

As shown in Figure 14, for Condition #1 and Condition #2, the vibration level differences from rail to track slab are basically the same due to the same type of fastener systems are employed. However, the vibration level differences of the two ballastless tracks from track slab to bridge deck are obviously different due to different types of isolation layers that are arranged underneath the track slabs. Regardless of the isolation layer types, the attenuation values and attenuation laws of vibrations in frequency range of 3000 to 5000 Hz are almost the same in these two conditions. Furthermore, the vibrations at higher frequency range of 3000 to 5000 Hz are attenuated rapidly with the distance along the vertical transmission path through the ballastless tracks. Under the conditions of different types of isolation layers, the vibration attenuation effect of frequency range of 0 to 3000 Hz is mainly affected, especially for frequency range of 1500 to 2000 Hz, the maximum level difference between Condition #1 and Condition #2 can reach 35 dB. It can be concluded that the vibration reduction and isolation effect of the rubber isolation layer is very significant compared with the geotextile isolation layer. As proved in practical tests, the introduction of rubber isolation layer in railway systems can effectively influence the vibration characteristics of ballastless track structures, leading to the reduction of vibrations dramatically.

## 6. Conclusions

In this work, we conduct a series of experimental research on the modal and vibration characteristics of unit-plate ballastless tracks used for high-speed railway long-span bridges. The main conclusions are as follows.

(1)The isolation layers change the vibration characteristics of track slab. The introduction of the rubber isolation layer in the ballastless track greatly reduces the vibration of track slab, and it also changes the vibration characteristics of track slab.(2)The isolation layers effect the vibration transmission characteristics in ballastless tracks. For the ballastless track with geotextile isolation layers or rubber isolation layers, the main difference of rail vibration is in frequency range of 2000 to 3000 Hz, and the reduction effects on rail vibration transmission caused by the rubber isolation layers are apparent. Moreover, the vibrations of the track slabs in ballastless tracks with geotextile or rubber isolation layers are reduced by 30% or 60%, which means that the vibration reduction effect of the rubber isolation layer is significant.(3)The application of rubber isolation layer makes the excited vibration frequency ranges of the ballastless tracks concentrated. Furthermore, the vibration attenuations of the ballastless tracks with rubber isolation layers are stable.(4)In the vertical vibration transmission process of the ballastless tracks with different types of isolation layers, the attenuation laws are slightly different in frequency domain. The vibration attenuation rate of the ballastless track with rubber isolation layers is about ten times larger than that of ballastless track with geotextile isolation layers, and the vibration reduction on vertical transmission in ballastless track with rubber isolation layers is apparent.

## Figures and Tables

**Figure 1 sensors-20-01744-f001:**
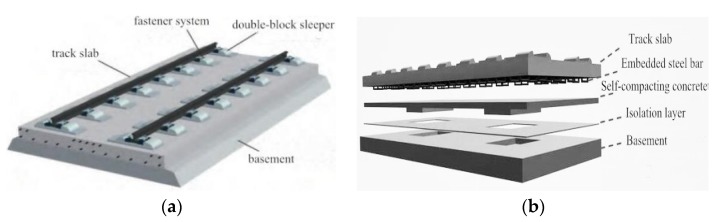
Unit-plate ballastless tracks. (**a**) Double-block type ballastless track; (**b**) CRTS Ⅲ type ballastless track.

**Figure 2 sensors-20-01744-f002:**
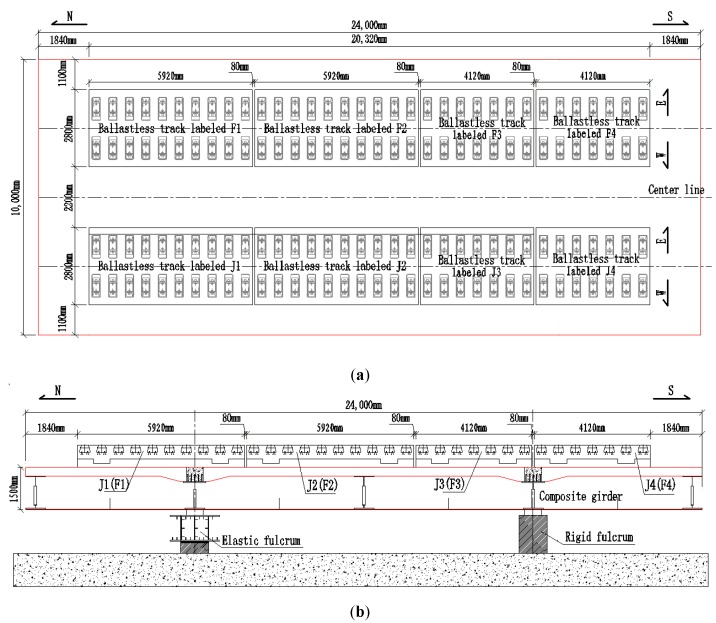
Full-scale unit-plate ballastless track used in this work. (**a**) Top view of the model; (**b**) side view of the model.

**Figure 3 sensors-20-01744-f003:**
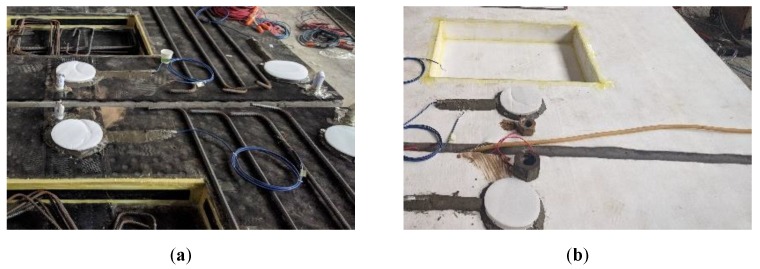
Different types of isolation layers in ballastless tracks. (**a**) Rubber isolation layer; (**b**) geotextile isolation layer.

**Figure 4 sensors-20-01744-f004:**
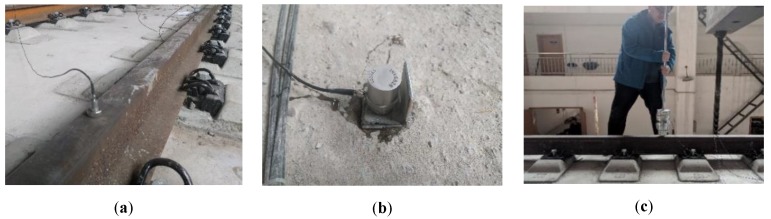
Sensors and hammering test. (**a**) The sensor on the rail; (**b**) The sensor on the concrete slab; (**c**) Hammering test.

**Figure 5 sensors-20-01744-f005:**
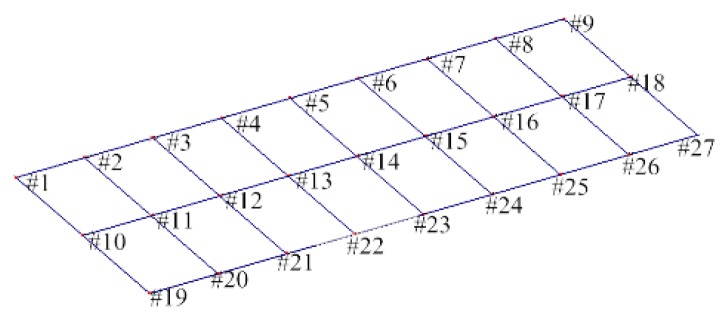
Layout of measuring points in modal measurement.

**Figure 6 sensors-20-01744-f006:**
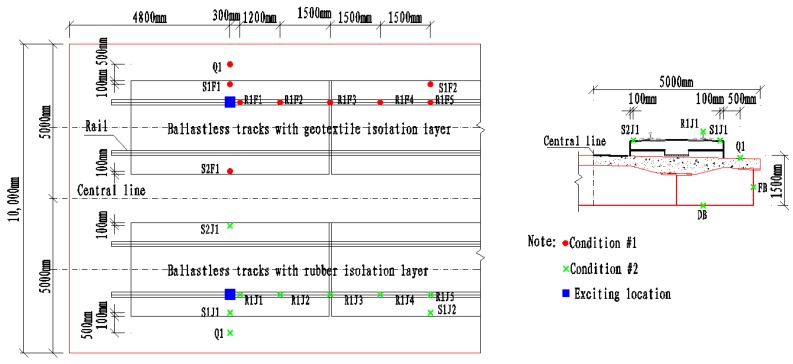
Arrangements of measuring sensors.

**Figure 7 sensors-20-01744-f007:**
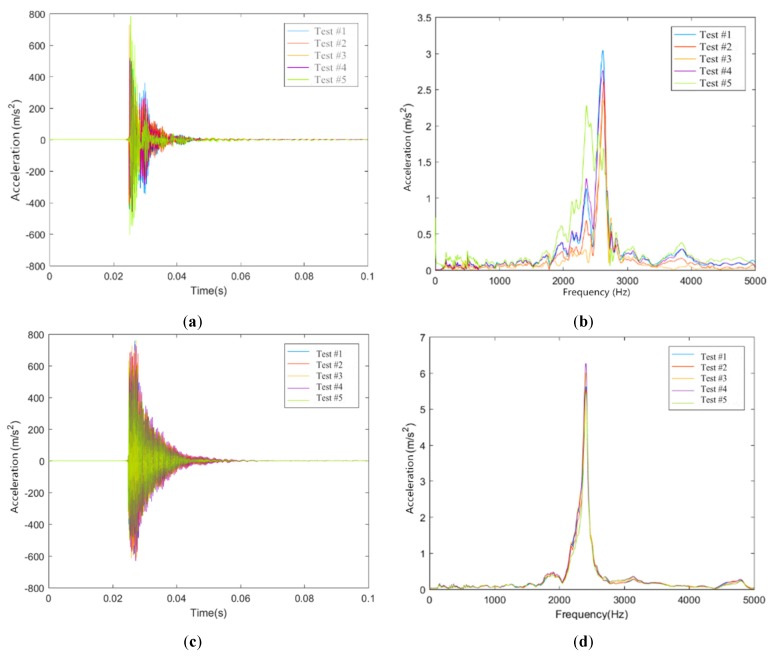
Vibration characteristics of rails. (**a**) Time-domain measuring data of sensor R1F1 in Condition #1; (**b**) Frequency-domain measuring data of sensor R1F1 in Condition #1; (**c**) Time-domain measuring data of sensor R1J1 in Condition #2; (**d**) Frequency-domain measuring data of sensor R1J1 in Condition #2.

**Figure 8 sensors-20-01744-f008:**
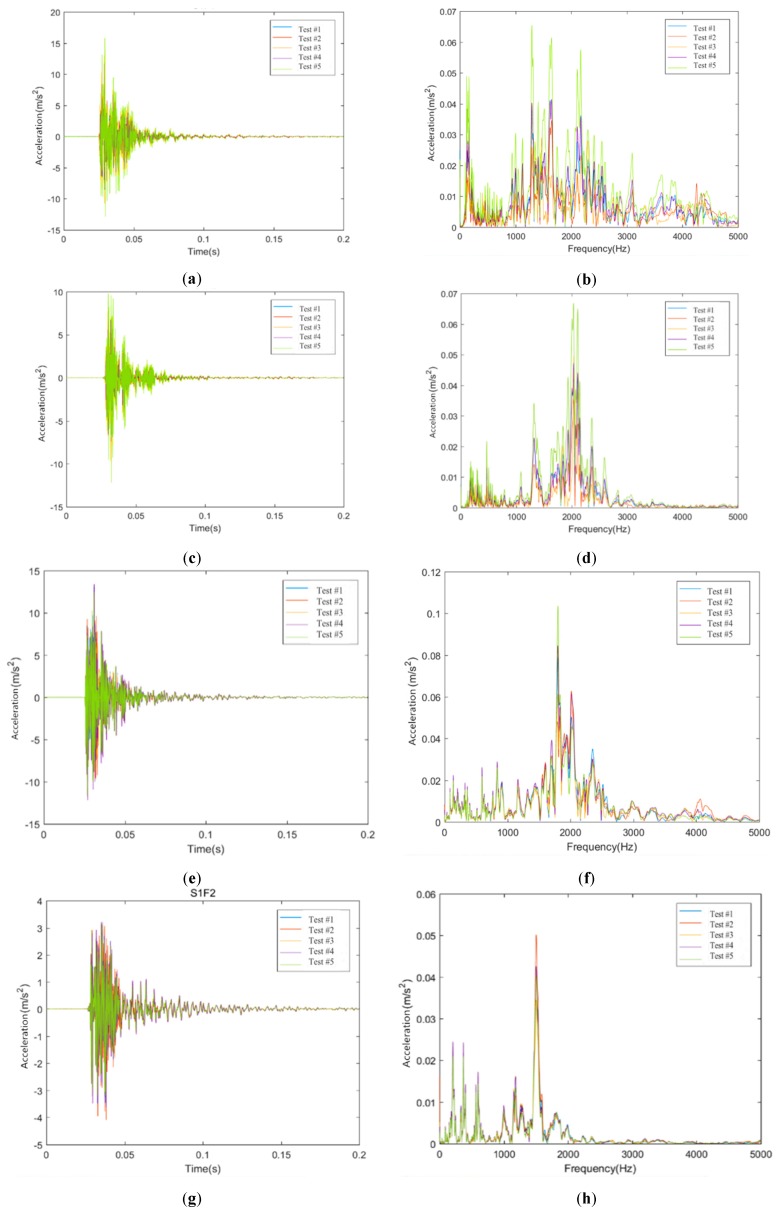
Vibration characteristics of track slabs. (**a**) Time-domain measuring data of sensor S1F1 in Condition #1; (**b**) Frequency-domain measuring data of sensor S1F1 in Condition #1; (**c**) Time-domain measuring data of sensor S1F2 in Condition #1; (**d**) Frequency-domain measuring data of sensor S1F2 in Condition #1; (**e**) Time-domain measuring data of sensor S1J1 in Condition #2; (**f**) Frequency-domain measuring data of sensor S1J1 in Condition #2; (**g**) Time-domain measuring data of sensor S1J2 in Condition #2; (**h**) Frequency-domain measuring data of sensor S1J2 in Condition #2.

**Figure 9 sensors-20-01744-f009:**
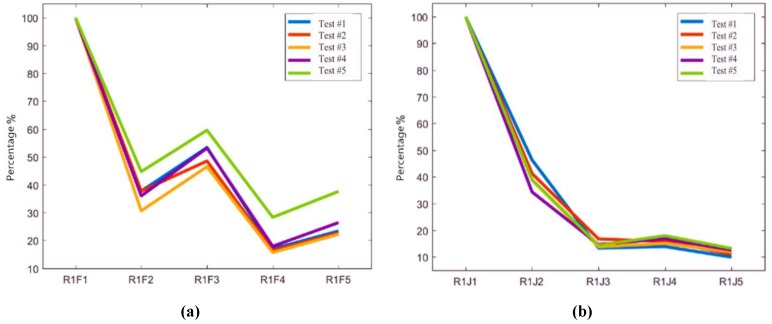
Longitudinal vibration transfer characteristics of rails. (**a**) Virtual vibration values of measuring sensors on rail in Condition #1; (**b**) Virtual vibration values of measuring sensors on rail in Condition #2.

**Figure 10 sensors-20-01744-f010:**
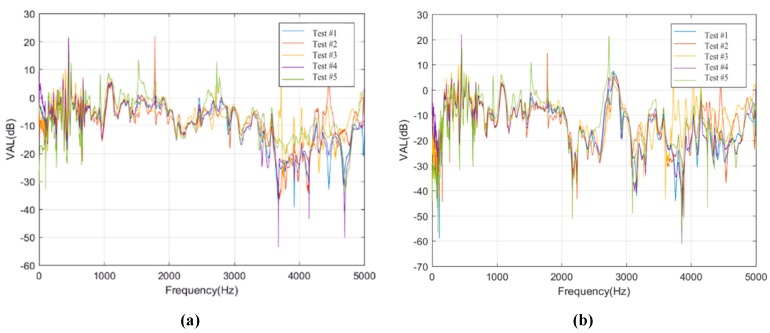
Acceleration level differences of the sensors on rails. (**a**) Acceleration level differences of sensor R1F3 in Condition #1; (**b**) Acceleration level differences of sensor R1F5 in Condition #1; (**c**) Acceleration level differences of sensor R1J3 in Condition #2; (**d**) Acceleration level differences of sensor R1J5 in Condition #2.

**Figure 11 sensors-20-01744-f011:**
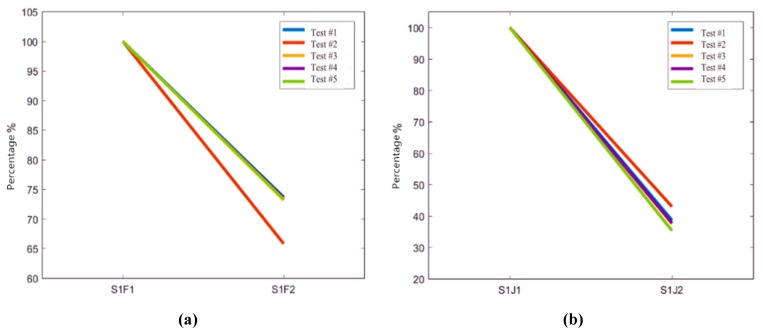
Vibration transfer rules of track slabs. (**a**) Virtual vibration value of measuring sensors on track slab in Condition #1; (**b**) Virtual vibration value of measuring sensors on track slab in Condition #2.

**Figure 12 sensors-20-01744-f012:**
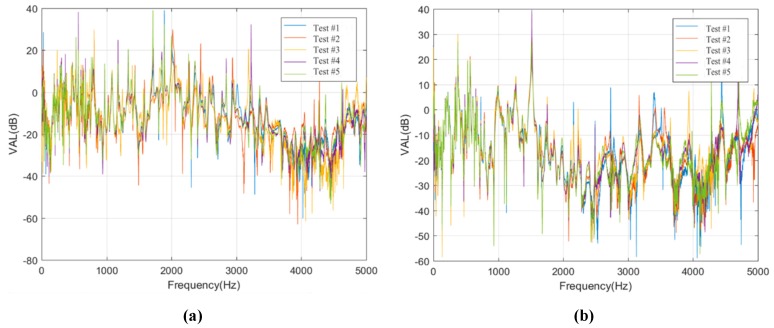
Acceleration level differences of the sensors on track slabs. (**a**) Acceleration level differences of sensor S1F2 in Condition #1; (**b**) Acceleration level differences of sensor S1J2 in Condition #2.

**Figure 13 sensors-20-01744-f013:**
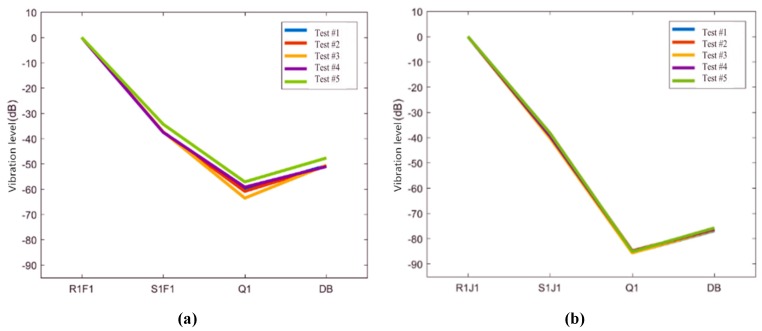
Vertical vibration transfer rules in ballastless tracks. (**a**) Virtual vibration value of the measuring sensors in Condition #1; (**b**) Virtual vibration value of the measuring sensors in Condition #2.

**Figure 14 sensors-20-01744-f014:**
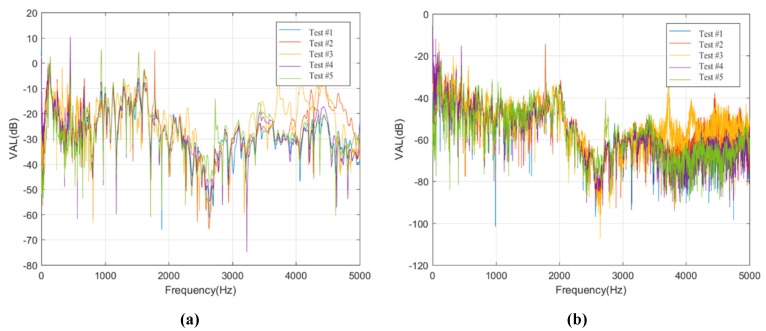
Acceleration level differences of different structural layers in ballastless tracks. (**a**) Acceleration level differences of sensor S1F1 in Condition #1; (**b**) Acceleration level differences of sensor Q1 in Condition #1; (**c**) Acceleration level differences of sensor S1J1 in Condition #2; (**d**) Acceleration level differences of sensor Q1 in Condition #2.

**Table 1 sensors-20-01744-t001:** Modal shapes of track slab in the two types of ballastless tracks.

Ballastless Track with Geotextile Isolation Layers		Ballastless Track with Rubber Isolation Layers	
First-order mode	Frequency (Hz)	First-order mode	Frequency (Hz)
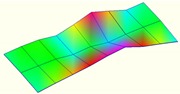	97.8	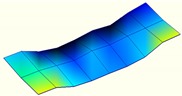	34.4
Second-order mode	Frequency (Hz)	Second-order mode	Frequency (Hz)
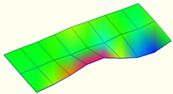	112.1	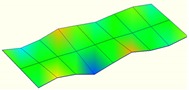	54.6
Third-order mode	Frequency (Hz)	Third-order mode	Frequency (Hz)
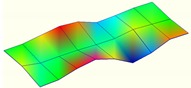	140.4	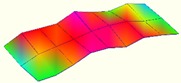	63.3

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
