# Peer review of "Experimental Study on Vibration Characteristics of Unit-Plate Ballastless Track Systems Laid on Long-Span Bridges Using Full-Scale Test Rigs"

_sensors, 2020, doi:10.3390/s20061744_

Round 1

Reviewer 1 Report

The main point of reviewed paper "Experimental Study on Vibration Characteristics of Unit-Plate Ballastless Track Systems Laid on Bridges Using Full-Scale Test Rigs" was that in my opinion it more technical paper than scientific. I didn't find scientific dimension. As a technical paper it well organized that is why gave positive decision.
The practical problem related to railway industry is solved. A lot experimental data provided. Used a lot well known methods of investigation. In the paper difficult to detect scientific novelty.
Also I am not sure is it fits to your journal politics accordingly to the main aim and scope.
In order to make paper better I would recommend to present some mathematical models, some simulations related to analyzed problems,
To organize and present methodology for generalized scheme of investigations and problem analysis.
Seeing in this way organized material in the paper it is appear meaning to give comment for paper improving.
I would like to repeat that like technical(engineering ) paper in my opinion is organized not so bad.

Author Response

Thank you for the reviewer’s comments. There is no denying that it is a new attempt to pave ballastless tracks on high-speed railway long-span bridges. The related issues deserve to be researched. Especially for the vibration characteristics of unit-plate ballastless track systems laid on the long-span bridges. Moreover, it is difficult to carry out experimental study based on full-scale test rigs. In this work, we conduct a series of experimental research on the modal and vibration characteristics of unit-plate ballastless tracks used for high-speed railway long-span bridge, and then some useful conclusions are obtained.

Reviewer 2 Report

I think the readers of this journal will appreciate the results of this manuscript.  Generally speaking, the material is judiciously divided and organized and correct from scientific point of view. The work is well structured, the presentation is judicious, the demonstrations are clear and the work is well written. The manuscript is well   organized. There experiments seem to be correct.

Some changes are, however, necessary. For these reasons I can recommend the acceptance of this paper after some corrections presented in the annexed file.

Author Response

Response to Reviewer 2 Comments

I think the readers of this journal will appreciate the results of this manuscript. Generally speaking, the material is judiciously divided and organized and correct from scientific point of view. The work is well structured, the presentation is judicious, the demonstrations are clear and the work is well written. The manuscript is well organized. There experiments seem to be correct. Some changes are, however, necessary. For these reasons I can recommend the acceptance of this paper after some corrections. Before that the Editor makes a decision, I suggest that the authors emphasize take into account the following corrections

Point 1: Please improve the Abstract with more explanations concerning the originality of the paper. In Abstract are presented some conclusions. Please move some conclusions to the section Conclusions.

Response 1: Thank you for the reviewer’s suggestions. We improve the section Abstract with more explanations concerning the originality of the paper, and move some sentences to the section Conclusions. All changes can be seen in the revised manuscript (section Abstract).

“Abstract:There is no denying that it is a new attempt to pave ballastless track on high-speed railway long-span bridges. The related issues deserve to be researched, especially for its vibration characteristics. In this work, we present a series of hammering tests on full-scale unit-plate ballastless tracks used for long-span bridges. The vibration characteristics and transmission rules of the ballastless track with geotextile or rubber isolation layers are explored, and the vibration reduction effect of the rubber isolation layer is analyzed. The main conclusions are as follows: The isolation layers change vibration modes and transmission characteristics of ballastless tracks. The introduction of rubber isolation layer makes the excited vibration frequency range of ballastless track concentrated, and the vibrations of ballastless track with rubber isolation layers are stable. Moreover, the rubber isolation layer has an obvious attenuation effect on vibration transmission in ballastless track structures. When vibration transmitted from rail to bridge deck, the vibration level differences of the ballastless track with rubber isolation layers are 20 dB larger than that of the ballastless track with geotextile isolation layers. The vibration attenuation rate of rubber isolation layer is about ten times larger than that of geotextile isolation layer”

Point 2: The section Conclusions will be point out the original results of the paper and can be extended to highlight the contributions.

Response 2: Thank you for the reviewer’s suggestions. We have tried our best to improve the section Conclusions with the aim of highlighting the contributions of our work. All revisions are marked in the revised manuscript.

Point 3: Some editing "glitches" need to be corrected.

Response 3: We have improved the manuscript and corrected some expressions. All revisions are marked in the revised manuscript.

Point 4: After each relationship a point, comma or semi-column should be placed.

Response 4: Thank you for the reviewer’s suggestions. We have revised the manuscript according to the reviewer’s comments. All revisions are marked in the revised manuscript.

Point 5: A very great number of notions and results are "borrowed" from different already published paper. As such, I think the authors need to emphasize more clearly the contribution of the manuscript from a scientific point of view;

Response 5: Thank you for the reviewer’s comments. We have tried our best to emphasize the contribution of the manuscript clearly. All revisions are marked in the revised manuscript.

Point 6: I think the authors need to emphasize more clearly the contribution of the manuscript from a scientific point of view;

Response 6: Thank you for the reviewer’s comments. We have tried our best to emphasize the contribution of the manuscript clearly. All revisions are marked in the revised manuscript.

Point 7: I propose a complete review of the paper in order to respect the Template.

Response 7: Thank you for the reviewer’s comments. We have tried our best to improve the manuscript.

Reviewer 3 Report

This work presents a series of impact tests based on full-size stepless slabs mounted on bridges. Vibration characteristics and principles of contactless track transmission with geotextile or rubber insulation layers were tested.
The vibration reduction effect of the rubber insulation layer is analyzed. The main conclusions are: Insulation layers change the vibration modes and the transfer characteristics of contactless track structures.

The introduction and description of the program test are legible and understandable.

Then in "3 Vibration characteristics of ballastless track systems" and "4 Vibration transmission characteristics of ballastless track systems" pictures 7 to 14 are presented. Test descriptions 1 to 5 appear there. Please provide a more detailed description of what test 1 is , test 2, test 3, test 4, test 5. There is nothing about this throughout the work.

In this work, Authors were present an investigation into the modal and vibration characteristics of the unit-plate ballastless track systems in high-speed railway bridge by conducting the hammering test on the full-scale test rigs. The conclusions resulting from the work are correct.

Work can be of great importance for the development of railways not only in China.

Author Response

This work presents a series of impact tests based on full-size stepless slabs mounted on bridges. Vibration characteristics and principles of contactless track transmission with geotextile or rubber insulation layers were tested.
The vibration reduction effect of the rubber insulation layer is analyzed. The main conclusions are: Insulation layers change the vibration modes and the transfer characteristics of contactless track structures.

Point 1: The introduction and description of the program test are legible and understandable.

Then in "3 Vibration characteristics of ballastless track systems" and "4 Vibration transmission characteristics of ballastless track systems" pictures 7 to 14 are presented. Test descriptions 1 to 5 appear there. Please provide a more detailed description of what test 1, test 2, test 3, test 4 and test 5 are. There is nothing about this throughout the work.

Response 1: We are so sorry that we failed to make us clearly. We added the detailed description of Test #1 to Test #5 in the revised manuscript (the second paragraph of section 1.2).

Point 2: In this work, Authors were present an investigation into the modal and vibration characteristics of the unit-plate ballastless track systems in high-speed railway bridge by conducting the hammering test on the full-scale test rigs. The conclusions resulting from the work are correct. Work can be of great importance for the development of railways not only in China.

Response 2: Thank you for the reviewer’s comments. We have improved the manuscript and corrected some expressions.

Round 2

Reviewer 2 Report

No comments